# Rapid and Accurate Identification of Nontuberculous Mycobacteria Directly from Positive Primary MGIT Cultures by MALDI-TOF MS

**DOI:** 10.3390/microorganisms10071447

**Published:** 2022-07-18

**Authors:** Laura Rindi, Vincenzo Puglisi, Iacopo Franconi, Roberta Fais, Antonella Lupetti

**Affiliations:** 1Dipartimento di Ricerca Traslazionale e delle Nuove Tecnologie in Medicina e Chirurgia, Università di Pisa, 56126 Pisa, Italy; aco89@live.it (I.F.); antonella.lupetti@unipi.it (A.L.); 2SD Microbiologia Micologica, Azienda Ospedaliero-Universitaria Pisana, 56126 Pisa, Italy; vincenzo.puglisi@ao-pisa.toscana.it (V.P.); roberta.fais91@live.it (R.F.)

**Keywords:** nontuberculous mycobacteria, MALDI-TOF MS, primary liquid culture

## Abstract

Over the last years, nontuberculous mycobacteria (NTM) have emerged as important human pathogens. Accurate and rapid mycobacterial species identification is needed to successfully diagnose, treat, and manage infections caused by NTM. Matrix-assisted laser desorption ionization time-of-flight mass spectrometry, MALDI-TOF MS, was demonstrated to effectively identify mycobacteria isolates subcultured from solid or liquid media rather than new positive cultures. The present study aims to develop a new extraction protocol to yield rapid and accurate identification of NTM from primary MGIT cultures by MALDI-TOF MS. A total of 60 positive MGIT broths were examined by the Bruker Biotyper system with Mycobacteria Library v. 2.0 (Bruker Daltonics GmbH & Co. KG., Bremen, Germany). The results were compared with those obtained by the molecular method, line probe assay GenoType Mycobacterium CM/AS/NTM-DR. All samples were concordantly identified by MALDI-TOF MS and the molecular test for all the tested mycobacteria. Fifty-seven (95%) MGIT positive cultures for NTM from clinical samples had a MALDI-TOF MS analysis score S ≥ 1.8. Although a small number of strains and a limited diversity of mycobacterial species were analysed, our results suggest that MALDI-TOF MS could represent a promising routine diagnostic tool for identifying mycobacterial species directly from primary liquid culture.

## 1. Introduction

Nontuberculous mycobacteria (NTM) are ubiquitous environmental microorganisms that can be recovered from soil and natural water as well as from drinking water sources, milk, and food products [1,2,3,4]. NTM are known to cause both asymptomatic and symptomatic infections in humans. Target sites of active symptomatic infection include mainly lung, followed by the central nervous system, lymph node, sinus, joint, and catheter-related infections [5,6]. With the exception of *Mycobacterium abscessus* among patients affected by Cystic Fibrosis (CF), no other NTM human-to-human or animal-to-human transmission has been demonstrated to date [7]. Acquisition of the microorganism is driven by environmental exposure through activities such as gardening, swimming, and showering [4]. The environmental burden of NTM required to induce symptomatic infection depends on the pre-existing host conditions. Patients affected by primary or secondary involvement of ciliary function are at higher risk of acquiring an NTM symptomatic infection [4,6]. Other non-respiratory conditions like gastro-oesophageal reflux disease, rheumatoid arthritis, cirrhosis, low levels of vitamin D, reduced body mass index (BMI), and malnutrition may affect NTM acquisition and subsequent development of diseases [4,6].

Immunosuppression is another major risk factor for NTM symptomatic infection [6]. Over the past decades, the rates of NTM infections have dramatically increased due to the rise in the number of immunosuppressed subjects and the development of more accurate identification tools [4,6]. Specifically, currently available diagnostic procedures include phenotypic and chemotaxonomic testing, genotypic methods using DNA-based molecular probes, high-performance liquid chromatography (HPLC), PCR, and DNA sequencing analysis. These analyses are time-consuming and must be performed in positive cultures. Phenotypic and chemotaxonomic testing is affected by specificity and sensitivity limitations [4,6]. Only DNA-based molecular probes and PCRs could also be run on direct sputum for NTM identification, but the data regarding their validation in clinical cohorts are still insufficient [8,9]. In our laboratory, for the identification of mycobacterial species, extraction, amplification, and hybridization with the Line Probe GenoType Mycobacterium CM test is performed on positive MGIT cultures in six hours and followed, if necessary, by a subsequent amplification and hybridization with the Line Probe GenoType Mycobacterium AS or NTM-DR test (Hain LifeScience, Nehren, Germany), which requires additional 5 h. The latter test is often performed the day after, depending on how the workflow is organized.

Matrix-assisted laser desorption ionization time-of-flight mass spectrometry (MALDI-TOF MS) proved to effectively identify mycobacteria isolates subcultured from solid media, such as Löwenstein–Jensen or Middlebrook 7H10 medium [10,11,12]. However, most studies did not apply MALDI-TOF MS for the identification of NTM directly from positive primary liquid cultures without a subculture step [13,14]. Subculture may require for NTM an average of 5 to 10 days depending on the isolated species. In consideration of hosts’ clinical pre-existing conditions and the severity of NTM diseases, accurate and rapid mycobacterial species identification on positive cultures is mandatory for successful diagnosis, treatment, and management of infections caused by NTM. Therefore, the aim of the present study was to develop a rapid method for direct identification of NTM from primary liquid cultures by MALDI-TOF MS.

## 2. Materials and Methods 

This is a prospective methodological study conducted at the Pisa University Hospital, Microbiology Unit, from February 2021 to January 2022.

***Clinical strains***: A total of 60 strains were isolated from clinical samples as bronchoalveolar lavage (BAL) fluid, sputum, ascitic fluid, pleural fluid, and skin samples were collected from patients undergoing investigation for a possible differential diagnosis with an NTM infection. All samples underwent direct microscopic examination performed with the Ziehl–Neelsen staining method, inoculation on both automated liquid culture systems (Mycobacteria Growth Indicator Tube (MGIT); BD Biosciences, Sparks, MD, USA), and solid medium, Löwenstein–Jensen agar, as part of current laboratory standard of care.

All positive MGIT broths were then examined by smear microscopy with the Ziehl–Neelsen staining method and underwent the Line Probe Assay GenoType Mycobacterium CM/AS/NTM-DR (Hain LifeScience, Nehren, Germany). In addition, MALDI-TOF MS analyses were performed on the same samples using the Bruker Biotyper system with Mycobacteria Library v 2.0 (Bruker Daltonics GmbH & Co. KG., Bremen, Germany).

***Internal reference strains***: Sixteen previously identified NTMs with Line Probe Assay GenoType Mycobacterium CM/AS/NTM-DR were collected from the NTM strain library within the Microbiology Unit at Pisa University. Next, each strain was inoculated on a MGIT liquid culture system and then analysed as internal reference strains following the same steps as for clinical strains.

***MALDI-TOF MS protein extraction protocol*:** Extraction was performed as soon as MGIT cultures became positive; we decided to perform this test within 24 h after automated growth detection. Protein extraction was carried out by the manufacturer’s MycoEx protocol [15]: mycobacterial biomass was resuspended in 300 µL H_2_O_2_. Next, absolute ethanol (900 µL) was added. The suspension was then centrifuged, and the ethanol was removed. The pellet was dried, and 0.5 mm zirconia/silica beads (40 µg) were added along with 25 µL acetonitrile. The pellet and beads were vortexed for 1 min, then 70% formic acid (25 µL) was added. After centrifugation at 13,000 rpm for 2 min, supernatant (1 µL) was deposed onto the MALDI-TOF MS plate.

An extract of *Escherichia coli* proteins (Bacterial Test Standard, BTS, Bruker Daltonics) was also deposed onto the MALDI-TOF MS plate as internal quality control, which is currently used in general bacteriology for MALDI-TOF MS calibration procedures.

MALDI-TOF MS results were compared with those obtained by the Line Probe Assay GenoType Mycobacterium CM/AS/NTM-DR (Hain LifeScience). Each sample was deposed onto three different spots on the MALDI-TOF MS plate; therefore, each sample had three independent scores. Scores were reported according to the Bruker system score value [12]. Specifically, analyses that showed results ≥1.8 were considered as ‘high confidence identification’, while scores ≥1.6 were considered as ‘low confidence identification’ [12]. 

Since clinical samples were taken as part of the standard patient care and MALDI-TOF MS analysis were performed on the remaining anonymized sample deposed as waste material, no written informed consent was necessary for this type of study.

## 3. Results

### 3.1. Identification of Clinical Strains

Sixty clinical samples were obtained: 31 BALs, 24 sputa, 3 pleuric fluids, 1 ascytic fluid, and 1 skin sample. The new protein extraction procedure was started within 24 h after the MGIT became positive according to MycoEx protocol and provided the mycobacterial species identification in less than 30 min. Our results showed that all identifications performed by MALDI-TOF MS gave interpretable results with different levels of confidence for all three replicate tests. Concordant identification between MALDI-TOF MS analysis and the NTM identification method currently used in our laboratory, Line Probe Assay GenoType Mycobacterium CM/AS/NTM-DR, was reported for all the mycobacteria isolated from both clinical and internal reference samples. In particular, for the clinical samples, MALDI-TOF MS analysis identified the following mycobacteria: *M. paragordonae/gordonae* (n. 16), *M. intracellulare/chimaera group* (n. 15), *M. avium* (n. 13)*, M. fortuitum* (n. 4), *M. kansasii* (n. 4), *M. abscessus* (n. 1), *M*. *mucogenicum/phocaicum* (n. 1), *M. xenopi* (n. 1), *M. chelonae* (n. 1), *M. celatum* (n. 1), *M. lentiflavum* (n. 1), *M. marinum* (n. 1), and *M. septicum* (n. 1), (Table 1).

Isolates were divided into three different groups according to the replicate scores, as shown in Table 1. The S ≥ 2.0 column had at least two out of three replicate scores ≥ 2.0; if two replicates showed a score of 1.8 ≤ S < 2, the sample was registered as 1.8 ≤ S < 2. For NTM obtained from clinical samples, 57 (95%) showed a ‘high confidence identification’ score ≥ 1.8. In particular, 51 MGIT positive cultures for NTM from clinical samples had a MALDI-TOF MS analysis score S ≥ 2.0 (85%); 6 had a MALDI-TOF MS analysis score 1.8 ≤ S < 2 (10%), while 3 (5%) had a MALDI-TOF MS analysis score of 1.6 ≤ S < 1.8.

MALDI-TOF MS cannot differentiate between *M. intracellulare* and *M. chimaera*—two closely related potentially pathogenic species of NTM that are members of the *M. avium* complex. Moreover, one clinical sample was identified as *M. fortuitum* complex by Line probe assay GenoType and *M. septicum* by MALDI-TOF MS. For this sample, results showed all three MALDI-TOF MS scores 1.6 ≤ S < 1.8, indicating a low confidence identification result. It is important to mention that *M. septicum* has been categorized phenotypically as a *Mycobacterium fortuitum* third biovariant complex member [16], thus explaining the different identification results.

### 3.2. Identification of Internal Reference Strains

In order to expand the number of NTM species, additional sixteen internal reference strains were analyzed: *M*. *intracellulare* (n. 2), *M*. *chimaera* (n. 1), M. *mucogenicum* (n. 1), *M*. *phocaicum* (n. 1)*, M. abscessus* (n. 2)*, M. avium* (n. 1)*, M. kansasii* (n. 1), *M. gordonae* (n. 1), *M. triplex* (n. 1)*, M. simiae* (n. 1)*, M. marinum* (n. 1), *M. parascrofulaceum* (n. 1)*, M. xenopi* (n. 1), *M. malmoense* (n. 1). All 16 isolates had a ‘high confidence identification’ result on MALDI-TOF MS analysis ≥ 1.8 (100%) for each replicate. The isolates were divided into three different groups according to the replicate scores, as shown in Table 2. Eleven isolates (69%) showed at least two out of three replicate scores ≥ 2; 5 samples (31%) showed all three replicates with a score of 1.8 ≤ S < 2.

The MALDI-TOF MS could not differentiate between *M. mucogenicum/phocaicum* (scores of S ≥ 2.0 for all three replicates). The 2 *M. mucogenicum/phocaicum* group strains were identified as *M. phocaicum* and *M. mucogenicum*, respectively, according to the GenoType Line Probe Assay.

## 4. Discussion

The present study shows how MALDI-TOF MS represents a useful and reliable diagnostic tool for the identification of mycobacterial species directly from primary liquid positive cultures. According to our results, all isolates were identified through MALDI-TOF MS technique showing a concordance of 100% with the method currently used in our laboratory, Line Probe Assay GenoType Mycobacterium CM/AS/NTM-DR analysis. A total of 60 positive MGIT cultures from clinical samples were examined, and 57 (95%) MGIT positive broths had a MALDI-TOF MS ‘high confidence identification’ score (S ≥ 1.8). The identification of mycobacterial species using the new protein extraction method took less than 30 min, thus proving to be extremely rapid. NTM species identified by MALDI-TOF MS included the species most frequently associated with human infections. All internal reference strains (16, 100%) reported a score ≥ 1.8 with the MALDI-TOF MS examination.

There are only a few studies that have evaluated the performance of MALDI-TOF MS for mycobacterial identification directly from primary liquid positive cultures. In a study by van Eck et al. [17], MALDI-TOF MS identification of NTM from primary cultures of respiratory samples yielded poor results, with 22% correct identification. This study disagrees with our results, although the platform used is the same, and the protein extraction protocol is very similar to ours. It is likely that the 30 min boiling step in the extraction procedure described by the authors reduces the performance of the identification system.

In a study by Kalaiarasan et al. that assessed MALDI-TOF VITEK MS using the sample preparation protocol provided by VITEK MS *Mycobacterium/Nocardia* kit (Biomerieux) on MGIT cultures, which were further incubated for three days before protein extraction, 52% of cultures were correctly identified [18]. A further study reported that the direct identification of NTM from positive MGIT broths, using MALDI-TOF VITEK MS with IVD v.3.0, generated high rates of acceptable results, reaching 96.4% and up to 100% (83/83) for sample preparations including a 0.1% SDS washing step [19]. Miller et al. [20] evaluated the accuracy of the Vitek MS v3.0 MALDI-TOF MS system, reporting that 87.7% of cultures were correctly identified; in this study, MGIT or VersaTREK liquid cultures were incubated for an additional 24 to 72 h before the protein extraction procedure. Finally, in a recent study, the MicroIDSys Elite system correctly identified 85.6% of the cultures with protein extraction that was performed 48–72 h after growth detection by MGIT [21]. Compared to these previous reports, the present study describes a protein extraction protocol applicable as soon as the MGIT culture becomes positive, i.e., within 24 h of culture positivity and 100% correct NTM identification, thus providing valuable information about the utility of MALDI-TOF MS for the rapid and accurate identification of NTM.

Our results should be interpreted with caution; due to the small number of samples analysed, no statistical analyses could be performed. Moreover, MALDI-TOF MS was unable to differentiate between *M. intracellulare* and *M. chimaera*, two closely related potentially pathogenic species of NTM that are members of the *M. avium* complex, as well as between *M. mucogenicum/phocaicum*, accounting for major study limitations. These strains were differentiated via the Line Probe Assay GenoType Mycobacterium CM assay followed by GenoType Mycobacterium NTM-DR assay the day after. 

Recently, a new MALDI-TOF MS subtyper software to distinguish *M. intracellulare* and *M. chimaera* based on different spectral peak signatures was developed, proving useful in *M. avium* complex species identification [22]. Despite study limitations, the concordance between MALDI-TOF MS results and the method currently used in our laboratory, the standardized extraction protocol procedure, and the reduced time required by MALDI-TOF MS (less than 30′) to perform the analysis are all relevant evidence that this technique could be implemented in current clinical practice. Our current clinical reference method requires at least six hours for the Line Probe Assay GenoType CM, and additional five hours if the Line Probe GenoType Mycobacterium AS or NTM-DR test must be performed. The latter may be tested the subsequent day, depending on how the workflow is organized. In comparison, the time required to run the MALDI-TOF analysis is less than 30 min. 

Another important point is related to the concept of time-saving: all the analyses were performed directly on positive MGIT cultures without any intermediate subculturing step, which may require an average of 5–10 days depending on the isolated NTM species. This, along with the reduced cost per analysis, could change the landscape of current laboratory diagnostics providing the clinician with a reliable result in a shorter period of time. Notably, the rapid identification of NTM could be of great usefulness in monitoring and controlling *M. abscessus* cross-transmission among CF patients [7]. In addition, MALDI-TOF MS is easy to perform and assess; therefore, our results can be externally validated and reproduced by other groups worldwide in future studies. This, along with the possibility of implementing a broader library spectrum able to differentiate between *M. intracellulare* and *M. chimaera,* as well as *M. mucogenicum* and *M. phocaicum* [23], makes us strongly believe that MALDI-TOF MS will become the preferred method of choice and reference diagnostic tool for NTM identification from primary liquid culture.

## Figures and Tables

**Table 1 microorganisms-10-01447-t001:** Identification of 60 nontuberculous mycobacterial species from clinical samples inoculated on MGIT liquid cultures by MALDI-TOF MS *.

Species (n.)	No. of Identified Isolates from Clinical Samples with Score (S) of:		
	1.6 ≤ S < 1.8	1.8 ≤ S < 2	S ≥ 2.0
*M. paragordonae/gordonae* (16)		2	14
*M. chimaera/intracellulare group* (15)		1	14
*M. avium* (13)		1	12
*M. fortuitum* (4)			4
*M. kansasii* (4)			4
*M. abscessus* (1)		1	
*M. mucogenicum/phocaicum* (1)			1
*M. xenopi* (1)			1
*M. chelonae* (1)	1		
*M. celatum* (1)		1	
*M. lentiflavum* (1)	1		
*M. marinum* (1)			1
*M. septicum* (1) ^§^	1		
Total 60 (%)	3 (5%)	6 (10%)	51 (85%)

* Using the results obtained with the routine methods by MALDI-TOF and GenoType Line Probe Assays as comparators. ^§^ Line Probe Assay GenoType analysis result for this clinical sample indicated *M. fortuitum.*

**Table 2 microorganisms-10-01447-t002:** Identification of 16 NTM from Internal reference strains inoculated on MGIT liquid cultures by MALDI-TOF MS *.

Species (n.)	No. of Identified Isolates from Internal Reference Strains with Score (S) of:		
	1.6 ≤ S < 1.8	1.8 ≤ S < 2	S ≥ 2.0
*M. chimaera/intracellulare group* (3)			3
*M. mucogenicum/phocaicum* (2) ^§^			2
*M. abscessus* (2) *M. avium* (1)		1	2
*M. kansasii* (1)		1	
*M. gordonae* (1)		1	
*M. triplex* (1)			1
*M. simiae* (1)		1	
*M. xenopi* (1)		1	
*M. malmoense* (1)			1
*M. marinum* (1)			1
*M. parascrofulaceum* (1)			1
Total 16 (%)	0 (0%)	5 (31%)	11 (69%)

* Using the results obtained with the routine methods by MALDI-TOF and GenoType Line Probe Assays as comparators. ^§^ Line Probe Assay GenoType analysis result for these clinical samples indicated 1 *M. mucogenicum* and 1 *M. phocaicum*.

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
