# Peer review of "Rapid and Accurate Identification of Nontuberculous Mycobacteria Directly from Positive Primary MGIT Cultures by MALDI-TOF MS"

_microorganisms, 2022, doi:10.3390/microorganisms10071447_

Round 1
Reviewer 1 Report
The manuscript titled: „Rapid identification of nontuberculous mycobacteria directly from positive primary MGIT cultures by MALDI-TOF MS” is interesting.
GENERAL COMMENTS
In my opinion, the research belongs to the pool of preliminary, basic. Unfortunately, they are not supported by a large number of repetitions and thus statistical analyzes. Despite the fact that the concept of the research and the first results are quite interesting, I find that they require broadening the analyzes, documenting them and thorough statistical analysis of the obtained data. The introduction is written at the appropriate substantive level. The methods are described correctly. As far as possible, the results are also presented accurately. In my opinion, the chapter on results should clearly emphasise the new procedures. The manuscript also lacks a study summary that should present the effectiveness of the introduced method. I hope that my tips will be useful in deciding the future of the manuscript. In my opinion, the research should be expanded with further results so that it can obtain the status of scientific basic research.
Author Response
Reviewer 1: The manuscript titled: „Rapid identification of nontuberculous mycobacteria directly from positive primary MGIT cultures by MALDI-TOF MS” is interesting.
We thank the reviewer for the thorough evaluation and improved our manuscript according to the suggestions below
Reviewer 1: In my opinion, the research belongs to the pool of preliminary, basic. Unfortunately, they are not supported by a large number of repetitions and thus statistical analyzes. Despite the fact that the concept of the research and the first results are quite interesting, I find that they require broadening the analyzes, documenting them and thorough statistical analysis of the obtained data.
Authors Response: We agree with the reviewer about the limitation of our manuscript, as we have pointed out in the Discussion section, lines 204-205: “…Our results should be interpreted with caution, due to the small number of samples analysed no statistical analysis could be performed....”. On the other hand, we believe that the protocol we have developed has provided very promising results, since this is the first paper showing a 100% concordance between MALDI-TOF MS technique and the Line Probe test. However, we were able to add 10 further samples analysed very recently, bringing the total number of positive MGIT cultures to 60 and we have edited the manuscript accordingly.
Reviewer 1: The introduction is written at the appropriate substantive level. The methods are described correctly. As far as possible, the results are also presented accurately.
Authors Response: We appreciate the Reviewer’s comments.
Reviewer 1: In my opinion, the chapter on results should clearly emphasise the new procedures.
Authors Response: As suggested, a sentence has been added to emphasize the new procedure (lines 117-119): “ The new protein extraction procedure was started within 24 h after the MGIT became positive according to MycoEx protocol and provided the mycobacterial species identification in less than 30 minutes.
Reviewer 1: The manuscript also lacks a study summary that should present the effectiveness of the introduced method.
Authors Response: The requested summary has been provided in the Discussion section (lines 172-179): “According to our results, all isolates were identified through MALDI-TOF MS technique showing a concordance of 100% with the method currently used in our laboratory, Line Probe assay GenoType Mycobacterium CM/AS/NTM-DR analysis. A total of 60 positive MGIT cultures from clinical samples were examined and 57 (95%) MGIT positive broths had a MALDI-TOF MS ‘high confidence identification’ score (S ≥ 1.8). The identification of mycobacterial species using the new protein extraction method took less than 30 minutes, proving extremely rapid. NTM species identified by MALDI-TOF MS included the species most frequently associated with human infections.”
Reviewer 2 Report
General Comments
1. There are a number of papers evaluating the use of MALDI-TOF against different methods for identification of nontuberculous mycobacteria and this manuscript does not clearly distinguish (if possible) its contribution from those published. Please distinguish this contribution to others.
2. The title includes "Rapid" as that is the unique contribution to available literature, but there is no side-by-side comparison between the methods used in the research. Please focus on "Rapid" if that is the intent of the authors.
3. A number of comments make identification of the value of the manuscript difficult, as "Rapid identification" is in the title, but later is added "develop a new extraction protocol to yield rapid and correct identification". Again, the reader asks what is the contribution?
Specific Comments
Abstract
Lines 20-23. The limited number of samples and therefore isolates limits the utility of any results.
Introduction
Line 44. What is meant by "...improvements in the detection methods." needs more explanation to identify the goals of the work.
Lines 57-64. Please distinguish the work in the manuscript with that available through publications.
Materials and Methods
Lines 86-103. AS there is note of an objective to "develop a new extraction protocol to yield rapid and correct identification of NTM..." what was the existing protocol and what metrics were employed to demonstrate "...rapid and correct..."?
Results
Tables 1 (lines 120-124) and 2 (lines 150-154). Particularly in light of the Sorin 3T heater-cooler-linked M. chimaera infections, a method that does not discriminate between M. intracellulare and M. chimaera is of little value. The same holds for discrimination of M. mucogenicum and M. phocaicam.
Discussion
Lines 169-177. Is the point that the work describes the value of MALDI following liquid culture? Is that where rapidity is increased? Again, the distinction between the published literature and this work needs to be made clearly.
Author Response
Reviewer 2: 1. There are a number of papers evaluating the use of MALDI-TOF against different methods for identification of nontuberculous mycobacteria and this manuscript does not clearly distinguish (if possible) its contribution from those published. Please distinguish this contribution to others.
Authors Response: We appreciate your constructive comment and we have addressed it in Introduction section (lines 59-64) and Discussion section (lines182-203). See below responses to specific comments.
Reviewer 2: 2. The title includes "Rapid" as that is the unique contribution to available literature, but there is no side-by-side comparison between the methods used in the research. Please focus on "Rapid" if that is the intent of the authors.
Authors Response: Compared to previous reports, the present study described a protein extraction protocol applicable within 24 h of culture positivity (lines 92-94) and 100% correct NTM identification. Therefore, we added “accurate” in the title and improved the discussion with a detailed comparison between methods available through publications (lines 182-203). Moreover, the text has been amended in Discussion section (lines 217-224): “Our current clinical reference method requires at least six hours for the Line Probe assay GenoType CM, and additional five hours if Line Probe GenoType Mycobacterium AS or NTM-DR test must be performed. The latter may be tested the subsequent day, depending on how the workflow is organized. In comparison, the time required to run MALDI-TOF analysis is less than 30 minutes. Another point that is important to stress, related to the time-saving concept, is that all the analysis was performed directly on positive MGIT cultures without any intermediate sub-culturing step, which may require an average of 5-10 days depending on the isolated NTM species”.
Reviewer 2: 3. A number of comments make identification of the value of the manuscript difficult, as "Rapid identification" is in the title, but later is added "develop a new extraction protocol to yield rapid and correct identification". Again, the reader asks what is the contribution?
Authors Response: We made the title and the text uniform by adding the term “accurate” (line 2; line 15), in line with the main intent of the manuscript which is represented by the description of a rapid and accurate identification method.
Abstract
Reviewer 2: Lines 20-23. The limited number of samples and therefore isolates limits the utility of any results.
Authors Response: We agree with the reviewer about the limitation of our manuscript and have therefore amended the last sentence in the Abstract (lines 21-24): “Although a small number of strains and a limited diversity of mycobacterial species were analysed, our results suggest that MALDI-TOF MS could represent a promising routine diagnostic tool for identification of mycobacterial species directly from primary liquid culture.” Moreover, we were able to add 10 further samples analysed very recently, bringing the total number of positive MGIT cultures to 60 and we have edited the manuscript accordingly.
Introduction
Reviewer 2: Line 44. What is meant by "...improvements in the detection methods." needs more explanation to identify the goals of the work.
Authors Response: In order to clarify this point, lines 45-46 has been changed as follow: “…development of more accurate identification tools [4,6]. Specifically, current available …”.
Reviewer 2: Lines 57-64. Please distinguish the work in the manuscript with that available through publications.
Authors Response: As requested, the last paragraph in the Introduction has been changed (lines 59-64): “Matrix-assisted laser desorption ionization time-of-flight mass spectrometry (MALDI-TOF MS) proved to effectively identify mycobacteria isolates sub-cultured from solid media, such as Lowenstein-Jensen or Middlebrook 7H10 medium [10-12]. However, most studies did not apply MALDI-TOF MS for identification of NTM directly from positive primary liquid cultures without a subculture step [13-14]. Subculture may require for NTM an average of 5 to 10 days depending on the isolated species.”
Materials and Methods
Reviewer 2: Lines 86-103. AS there is note of an objective to "develop a new extraction protocol to yield rapid and correct identification of NTM..." what was the existing protocol and what metrics were employed to demonstrate "...rapid and correct..."?
Authors Response: There is no unique reference protocol. Only few studies have evaluated MALDI-TOF identification of NTM directly from positive liquid cultures and have proposed different protocols for protein extraction as well as different MALDI-TOF MS platform and database. Comments on this aspect have been added in the Discussion section (lines 182-203).
Results
Reviewer 2: Tables 1 (lines 120-124) and 2 (lines 150-154). Particularly in light of the Sorin 3T heater-cooler-linked M. chimaera infections, a method that does not discriminate between M. intracellulare and M. chimaera is of little value. The same holds for discrimination of M. mucogenicum and M. phocaicam.
Authors Response: We agree with the reviewer about the limitation of the discrimination power of the method regarding M. intracellulare/M. chimaera and M. mucogenicum/M. phocaicum, as we have pointed out in the Discussion section, lines 205-208: “Moreover, MALDI-TOF MS was unable to differentiate between M. intracellulare and M. chimaera, two closely related potentially pathogenic species of NTM that are members of the M. avium complex as well as between M. mucogenicum/phocaicum, accounting for major study limitations.”. However, the implementation of the updated version of the Bruker Biotyper system, that could resolve this problem, will be acquired in the next future.
Discussion
Reviewer 2: Lines 169-177. Is the point that the work describes the value of MALDI following liquid culture? Is that where rapidity is increased? Again, the distinction between the published literature and this work needs to be made clearly.
Authors Response: In order to clarify the distinction between the published literature and this work the paragraph of Discussion section (lines 182-203) has been extended and deepened: “In a study by van Eck et al. [17] MALDI-TOF MS identification of NTM from primary cultures of respiratory samples yielded poor results with 22% correct identification. This study disagrees with our results, although the platform used is the same and the protein extraction protocol is very similar to ours; probably the 30 min boiling step in the extraction procedure described by the authors reduces the performance of the identification system. In a study by Kalaiarasan et al., that assessed MALDI-TOF VITEK MS using the sample preparation protocol provided by VITEK MS Mycobacterium/Nocardia kit (Biomerieux) on MGIT cultures which were further incubated for three days before protein extraction, 52% of cultures were correctly identified [18]. A further study reported that direct identification of NTM from positive MGIT broths, using MALDI-TOF VITEK MS with IVD v.3.0, generated high rates of acceptable results reaching 96.4%, and up to 100% (83/83) for sample preparations including a 0.1% SDS washing step [19]. Miller et al [20] evaluated the accuracy of the Vitek MS v3.0 MALDI-TOF MS system reporting 87.7% cultures correctly identified; in such study MGIT or VersaTREK liquid cultures were incubated for an additional 24 to 72 h before protein extraction procedure. Finally, in a last recent study, the MicroIDSys Elite system correctly identified 85.6%, with protein extraction that was performed 48-72 h after growth detection by MGIT [21]. Compared to these previous reports, the present study described a protein extraction protocol applicable as soon as MGIT culture becomes positive, i.e. within 24 h of culture positivity and 100% correct NTM identification, thus providing valuable information about the utility of MALDI-TOF MS for the rapid and accurate identification of NTM.”
Round 2
Reviewer 1 Report
In my opinion, the communication titled: „Rapid identification of nontuberculous mycobacteria directly from positive primary MGIT cultures by MALDI-TOF MS” is interesting. After considering all reviewers' comments, its substantive value increased significantly.
The communication’s results are reproducible based on the details given in the methods section. The data was interpreted appropriately and consistently throughout the manuscript. The discussion chapter is corrected and enriched with new content.
Thank you for taking into account all comments and suggestions of the reviewer.
Reviewer 2 Report
The revised manuscript is much improved and acceptable for ublication.